# Influence of the Team Effectiveness of Nursing Units on Nursing Care Left Undone and Nurse-Reported Quality of Care

**DOI:** 10.3390/healthcare11101380

**Published:** 2023-05-10

**Authors:** Se Young Kim, Young Ko

**Affiliations:** 1Department of Nursing, Changwon National University, 20 Changwondaehak-ro, Uichang-gu, Changwon 51140, Republic of Korea; sarakimk@changwon.ac.kr; 2College of Nursing, Gachon University, Incheon 21936, Republic of Korea

**Keywords:** team effectiveness, nursing unit, care left undone, incomplete care, quality of care

## Abstract

The aim of this study was to identify the influence of nursing unit team effectiveness on nursing care left undone and nurse-reported quality of care. This was a cross-sectional study with a sample of 230 nurses working at general hospitals in South Korea. Data were collected in January 2023 using an online questionnaire. Nursing unit team effectiveness was measured, consisting of the following sub-scales: leadership of the head nurse, cohesion, job satisfaction, competency of nurses, work productivity, and coordination. Multiple regression analyses were used to assess relationships between nursing unit team effectiveness and nursing care left undone and nursing-reported quality of care. Among these sub-domains, the study found that the higher the coordination (β = −0.22, *p* < 0.001), the significantly lower the nursing care left undone. The higher the competency of nurses (β = 0.26, *p* < 0.001) and work productivity (β = 0.20, *p* < 0.001), the higher the nurse-reported quality of care. In addition, nursing care left undone had a negative effect on nurse-reported quality of care (β = −0.15, *p* < 0.001). Therefore, nursing managers should make efforts to manage team effectiveness in nursing units to improve nurse-reported quality of care.

## 1. Introduction

The Institute of Medicine’s “To Err is Human: Building a Safer Health System” report [1] revealed that effective teamwork and improved communication among healthcare providers can prevent half of adverse events. In addition, “to promote effective team functioning” was described as one of the five principles for developing a safe hospital system. As patient safety continues to gain in importance, there has been a growing focus on studying team effectiveness in healthcare [2].

Cohen and Bailey define a team as “A collection of individuals who are interdependent in their tasks, who share responsibility for outcomes, who see themselves and who are seen by others as an intact social entity embedded in one or more larger social systems (for example, business unit or corporation), and who manage their relationships across organizational boundaries” [3]. Several models have been developed to conceptualize the aspects that influence team effectiveness [2,4,5,6].

According to the Integrated Team Effectiveness Model for team effectiveness in healthcare organizations, organizational context and task design affect team effectiveness through team processes and team psychosocial traits [4]. The effectiveness of multidisciplinary teams in healthcare organizations, with a focus on doctors, has been evaluated using objective variables such as changes in patient health status, patient satisfaction, survival rate, mortality, risk events, length of hospital stay in days, and cost-effectiveness of healthcare services [4].

Kozlowski and Bell defined work teams as “collectives who exist to perform organizationally relevant tasks, share one or more common goals, interact socially, exhibit task interdependencies, maintain and manage boundaries, and are embedded in an organizational context that sets boundaries, constrains the team, and influences exchanges with other units in the broader entity” [7].

A nursing unit can be described as a work team that provides direct and indirect nursing care, supportive nursing, and communication functions to a specific number of patients [8]. In the nursing unit, nurses cooperate through shift work and handovers to implement nursing plans according to the nursing process. Nursing units are considered work teams that share a common goal and are assembled to deliver nursing care to patients [9,10]. Therefore, it is necessary to measure and understand the team effectiveness of nursing units [8].

In addition, process factors such as coordination and communication between members and multi-dimensional characteristics of the health care team should be reflected when measuring the team effectiveness of nursing units [4,8]. Accordingly, Kim and Kim [11] developed the Team Effectiveness Scale for Nursing Units (TES-NU), based on the Integrated Team Effectiveness Model. The TES-NU consists of six sub-scales: head nurse leadership, cohesion, job satisfaction, nurse competency, work productivity, and coordination. Therefore, it is necessary to measure team effectiveness as a performance measure of the nursing unit by distinguishing it from the performance of a multi-disciplinary healthcare team and identifying the outcomes of team effectiveness [12].

Nurse-reported quality of care is a useful indicator of hospital outcome evaluation, as it is significantly related to patient mortality, patient satisfaction, acute myocardial infarction, pneumonia, and surgical patient outcomes [13]. Nurses are present in all situations where patient care is provided in hospitals, making them accessible and convenient sources of information. Several studies have used nurse-reported quality of care to measure health care outcomes [13,14,15,16,17]. Nurses’ assessments of patient–provider interactions, patient–guardian education and support, professional collaboration, technology and information systems implementation, and staff management are not always documented in medical records but can impact healthcare outcomes. Therefore, nurse-reported quality of care is a valid indicator of patient outcomes and nursing processes [13]. In addition, it is predicted that nurses will have a higher perception of the quality of care when the team effectively achieves this goal. In this context, it is necessary to measure nurse-reported quality of care as an outcome variable of the team effectiveness of nursing units and to understand its relevance.

Nursing care left undone refers to nursing care that fails to perform necessary duties, leading to unmet patient needs. It is often observed in work environments with a large nursing workload or insufficient nursing resources, such as human resources and time [18,19]. Between 55% and 98% of nurses have reported not being able to finish their work during working hours [20], and an average of 4.4 out of 13 nurses were found to have failed to complete nursing tasks during working hours [21]. Nursing care left undone can lead to negative patient outcomes such as falls, mishandling of medication, errors related to procedures or treatments, bedsores, and hospital infections, ultimately reducing the quality of nursing care [21]. Previous studies have found that through interventions to improve team effectiveness, the speed and completion of work in the emergency room improved [22]. Such improvement was achieved through team training [23,24,25]. Therefore, it is expected that the occurrence of nursing care left undone would decrease if nurses received adequate training on how to effectively share and coordinate tasks in the nursing unit and positively perceive the work performance of the nursing unit [11]. It is necessary to empirically confirm there is less nursing care left undone in nursing units with a high team effectiveness. By confirming the relationship between nursing care left undone and team effectiveness, team effectiveness in nursing units can be used as a strategy to reduce nursing care left undone.

So far, only a few studies have investigated the relationship between team effectiveness in nursing units and the quality of nursing care. Therefore, this study aims to identify the level of team effectiveness in nursing units by looking at the six sub-scales and examining their relationships with nursing care left undone and nurse-reported quality of care. Overall, the goal is to highlight and provide basic data for the improvement of team effectiveness in nursing organizations.

## 2. Materials and Methods

### 2.1. Study Design and Participants

This study employed a cross-sectional survey design to examine the influence of team effectiveness in nursing units on nursing care left undone and nursing quality. Participants were nurses working in three general hospitals in Korea. Convenience sampling was used to select the hospitals and nurses. G*Power 3.1 was used to calculate the number of samples needed for multiple regression analysis based on a significance level of 0.05, explanatory power of 0.95, medium effect size of 0.15, and 20 predictor variables. The minimum sample size for multiple regressions was predicted to be 225, and so 250 nurses were recruited to account for an anticipated dropout rate of 10%. Ultimately, a total of 230 nurses responded to the survey, and their data were included in the analysis.

### 2.2. Measures

#### 2.2.1. Team Effectiveness of Nursing Unit

For this study, we used the Team Effectiveness Scale for Nursing Units (TES-NU), developed by Kim and Kim [11] based on the Integrated Team Effectiveness Model [4]. This tool consists of 30 items across six subscales, with each item being measured using a 5-point Likert scale, and higher scores indicating higher team effectiveness in the nursing unit. The tool’s reliability was assessed using Cronbach’s alpha, which was 0.94 overall. Meanwhile, Cronbach’s alpha values for each sub-scale were: “leadership of the head nurse”, 0.90; “cohesion”, 0.88; “job satisfaction”, 0.87; “competency of nurses”, 0.82; “work productivity”, 0.79; and “coordination” 0.70 [4]. In this study, the overall Cronbach’s alpha value was 0.96, with the following values for each sub-scale: “leadership of the head nurse”, 0.92; “cohesion”, 0.89; “job satisfaction”, 0.87; “competency of nurses”, 0.82; “work productivity”, 0.95; and “coordination” 0.71. 

#### 2.2.2. Nursing Care Left Undone

In this study, we used the Nursing Care Left Undone tool developed by Ausserhofer et al. [18] and translated by Park and Hwang [21]. The tool includes thirteen items that assess nursing activities left undone during recent work periods. The items include “comfort/talk with patients”, “developing or updating nursing care plans/care pathways”, “educating patients and families”, “oral hygiene”, “adequate patient surveillance”, “planning care”, “frequent changing of patient position”, “skin care”, “preparing patients and families for discharge”, “administering medications on time”, “pain management”, and “treatments and procedures”. Each item is scored with 1 point for “yes” and 0 points for “no” and the scores are summed up to yield a final score ranging from 0 to 13 points, with higher scores indicating more nursing care left undone. In the study by Park and Hwang [21], KR-20 was 0.86, whereas in this study it was 0.70.

#### 2.2.3. Nurse-Reported Quality of Care

We measured nurse-reported quality of care (NQoC) using the following question: “How would you describe the quality of nursing generally provided to patients?” Participants rated their responses on a 5-point Likert scale ranging from 1 (very low) to 5 (very excellent), with higher scores indicating higher perceived quality of patient care. Quality of care reported by nurses has been widely used in various international studies, and the validity of using a single question has been previously established [13,14].

#### 2.2.4. Covariates

We considered covariates such as sex, age, education, marital status, nursing experience, nursing experience in their current unit, position, type of nursing unit, and number of beds per total number of nurses in nursing unit. Number of beds per total number of nurses in nursing unit is calculated as the number of beds divided by the number of full-time nurses in nursing unit [22]. 

### 2.3. Data Collection

Data were collected in January 2023. This study was conducted with the approval of the IRB of Changwon National University (IRB no: 7001066-202209-HR-064). The researchers visited three general hospitals and explained the purpose and methods of the study. Participant recruitment was announced by the nursing department of each hospital. Data were collected through an online questionnaire from those who voluntarily agreed to participate after reading the purpose and methods of the study. 

### 2.4. Statistical Analysis

We analyzed the data using the Stata 16.0 statistical program (Stata Corp., College Station, TX, USA). Participants’ characteristics, level of nursing unit team effectiveness, nursing care left undone, and nurse-reported quality of care were presented as frequencies and percentages, averages and standard deviations. Pearson’s correlation was used to analyze the relationships between nursing unit team effectiveness, nursing care left undone, and nurse-reported quality of care. We conducted multiple regression analyses, with team effectiveness in nursing units and covariates (nursing experience in the current unit, position, type of nursing unit) as independent variables, with nursing care left undone and nurse-reported quality of care as the dependent variables. Participants’ general and job-related characteristics that were significantly related to team effectiveness, nursing care left undone, and the quality of nursing care were entered into the multiple regression model. 

## 3. Results

### 3.1. Participant Characteristics

Participants’ characteristics are presented in Table 1. Approximately 93% of participants were female and the average age was 29.90 (±5.73) years old. Approximately 66% of participants had a bachelor’s degree in nursing and 75.2% were unmarried. The average length of nursing experience was 6.18 (±5.59) years, and the average length of nursing experience in their current unit was 3.74 (±3.69) years. The average number of beds per total number of nurses in nursing unit was 2.22 (±0.68). Approximately 66% of participants were general nurses, 65.7% worked in a general nursing care service ward, and 34.4% worked in an integrated nursing care service ward.

### 3.2. Team Effectiveness, Nursing Care Left Undone, and Quality of Nursing

Table 2 shows the level of team effectiveness of the nursing unit, nursing care left undone, and nurse-reported quality of care. The average total score of team effectiveness was 3.80 (±0.55) out of 5 points. Among the sub-scales of team effectiveness, leadership of the head nurse and the competency of nurses received the highest scores, of 3.94 (±0.78) and 3.94 (±0.64), respectively. Cohesion, work productivity, job satisfaction, and coordination received lower scores. The average score for nursing care left undone was 1.43 (±1.78) points out of a possible range of 0–13 points, while the average score for nurse-reported quality of care was 3.75 (±0.71) points out of 5. There was no statistically significant difference in team effectiveness based on the general and job-related characteristics of the participants. However, job satisfaction, as a subscale of team effectiveness, showed significant differences according to age, nursing experience, and nursing experience in the current unit. With the exception of the type of nursing unit, no general or job-related characteristics were related to nursing care left undone. There was no significant difference in the nurse-reported quality of care according to general and job-related characteristics. 

Table 3 presents the results of the analysis of the relationships among team effectiveness, nursing care left undone, and nurse-reported quality of care. Team effectiveness showed a negative correlation with nursing care left undone (r = −0.23, *p* < 0.001) and a positive correlation with the quality of nursing (r = 0.59, *p* < 0.001). Meanwhile, there was a negative correlation between nursing care left undone and quality of care (r = −0.31, *p* < 0.001).

### 3.3. Influence of Team Effectiveness of Nursing Unit on Nursing Care Left Undone and Nurse-Reported Quality of Care

Table 4 shows the influence of nursing unit team effectiveness on nursing care left undone and nurse-reported quality of care. In the regression model, nursing experience and the type of nursing unit (which are characteristics related to team effectiveness), nursing care left undone, and quality of nursing were entered. The goodness-of-fit of the model and multi-collinearity among the independent variables were assessed and all assumptions of the regression analysis were satisfied.

After analyzing the influence of the team effectiveness of the nursing unit on nursing care left undone, we found that higher levels of team coordination (β = −0.22, *p* < 0.001) and providing integrated nursing care service (β = −0.17, *p* < 0.001) were significantly associated with less nursing care left undone.

Moreover, an analysis of the influence of the team effectiveness of the nursing unit on nurse-reported quality of care showed that higher nurse competency (β = 0.26, *p* < 0.001) and work productivity (β = 0.20, *p* < 0.001), along with providing an integrated nursing care service (β = 0.12, *p* < 0.05) and lower levels of nursing care left undone (β = −0.15, *p* < 0.001), were significantly associated with better nurse-reported quality of care.

## 4. Discussion

The primary unit of a nursing organization, known as the nursing unit, is a self-controlled and individual production unit that provides professional nursing care services according to health status and nursing needs of a certain number of patients [23]. In this study, we aimed to investigate the team effectiveness of nursing units and its influence on nursing care left undone and nurse-reported quality of care.

Our finding showed that nursing care left undone decreased as the level of “coordination” in the team effectiveness of the nursing unit increased. Coordination was measured by assessing the predictability of the work perceived by nurses, the appropriateness of work sharing, and the degree of doctors’ confidence in nurses’ expertise. This subscale corresponds to the team process factor that affects team effectiveness in the ITEM model. Therefore, these results may reflect a decrease in the occurrence of nursing care left undone when tasks are predictable and properly shared within the nursing unit, supporting the hypothesis that team process has an impact on team effectiveness in the ITEM model. The leading causes of nursing care left undone, which negatively affect patient safety, were found to be an excessive workload and inadequate staffing [18,19]. In addition, communication and teamwork between nurses, between nurses and doctors, and between nurses and assistants in the nursing unit have been previously identified as factors influencing the occurrence of nursing care left undone. Internal factors of nursing units, such as team norms, have also been found to be related to nursing care left undone [24]. Each team has a unique set of norms, and new members quickly learn what is acceptable behavior within the team. For example, it has been shown that if nurses neglect prescribed patient care or discharge plans, or if senior nurses skip certain tasks, newly hired nurses tend to approve of such behavior. This tendency appears because they know that they will be ostracized by the more experienced nurses if they do not comply with these norms [25]. Additional research is needed to further understand the norms of nursing units and to develop new strategies for identifying predictable tasks, managing tasks properly among nurses, and improving communication among members of the nursing unit. Nursing organizations should improve staffing levels to reduce the nurse-to-patient ratio and thus help reduce the occurrence of nursing care left undone.

In this study, the factors influencing nurse-reported quality of care were found to be “work productivity” and “competency of nurses”, which are sub-scales of team effectiveness in the nursing unit. The ITEM model [4] measures team effectiveness in nursing units using two factors: “competence of nurses”, which corresponds to team composition and reflects a person’s necessary qualities, expertise, and responsibility for team effectiveness, and “work productivity”, which is a subjective outcome of team effectiveness and is measured by the effectiveness, efficiency, and error-free completion of tasks by fellow nurses to achieve a common goal [11]. This result aligns with the findings of a previous study that showed the importance of nurses’ recognition of their colleagues’ competence in achieving the goals of the nursing unit [11]. Nurses who have the necessary qualifications, expertise, and responsibility to perform their duties well seem to evaluate the quality of nursing care positively. The results of this study are consistent with previous research that suggests that a team’s cognitive ability predicts its performance and that teams with high-level task knowledge perform better, especially when carrying out intellectual and decision-making tasks [12,26]. Further, the productivity of the nursing unit is closely tied to the unit’s work process, which is often carried out through shift work and handovers. As described above, this study confirmed that the “competence of nurses” and “work productivity” of team effectiveness are significant factors in determining nurse-reported quality of care as a result of team effectiveness [12,13]. These findings indicate that nurses perceive the quality of nursing provided by the unit more positively when their fellow nurses possess the necessary capabilities to perform their duties and can achieve common goals effectively and efficiently.

In a study by Buljac-Samardzic et al. [27], which analyzed interventions aimed at enhancing team effectiveness in healthcare, it was reported that crew resource management (CRM) and TeamSTEPPS training were mainly applied to improve teamwork and patient safety in hospitals, but these interventions improved only non-technical skills such as teamwork, communication, and safety culture. According to a previous study, simulation-based training developed from clinical scenarios can improve both technical and/or non-technical skills [28]. Therefore, to increase team effectiveness in nursing units, nursing organizations need to provide CRM and TeamSTEPPS training to improve teamwork and communication, as well as simulation training to strengthen task knowledge [12,26] and practical skills. 

This study’s strength is the identification of sub-units of nursing team effectiveness that affect nursing care left undone and overall quality of nursing care. However, it should be noted that this study employed a cross-sectional design and convenience sampling, and as such, the results should be interpreted with caution and consideration of the limitations in inferring causality. As an objective measure of nursing unit team effectiveness, future studies may consider measuring and confirming the relevance of patient functional status, satisfaction, externally evaluated nursing quality, and organizational cost-effectiveness. Further, this study did not investigate differences in team effectiveness based on nursing delivery methods such as functional nursing, team nursing, primary nursing, and modular nursing, or types of wards such as general wards and integrated nursing care service. Therefore, we suggest further research to identify differences in team effectiveness according to type of nursing unit and care delivery method.

## 5. Conclusions

By examining the relationship between team effectiveness, nursing care left undone, and nurse-reported quality of care, this study confirmed that coordination of team effectiveness in nursing units could reduce nursing care left undone. Moreover, nurses’ competency and work productivity were found to have a positive effect on nurse-reported quality of care. Therefore, nursing management should prioritize efforts to increase team effectiveness in nursing units to improve the quality of care. Lastly, coordinating work in the nursing unit to reduce nursing care left undone can further enhance the quality of nursing care.

## Figures and Tables

**Table 1 healthcare-11-01380-t001:** Participants’ characteristics (*n* = 230).

Variables	Category	*n* (%) or m ± SD
Sex	Male	17 (7.4)
	Female	213 (92.6)
Age (years)		29.90 ± 5.73
Education	Diploma	60 (26.1)
	Bachelor’s	152 (66.1)
	Master’s	18 (7.8)
Marital status	Unmarried/separated	173 (75.2)
	Married/partnered	57 (24.8)
Nursing experience (years)		6.18 ± 5.59
Nursing experience in their current unit (years)		3.74 ± 3.69
Position	Staff nurse	214 (93.0)
	Charge nurse	16 (7.0)
Type of nursing unit	General nursing care service	151 (65.7)
	Integrated nursing care service	79 (34.4)
Number of beds per total number of nurses in nursing unit		2.22 ± 0.68

**Table 2 healthcare-11-01380-t002:** Team effectiveness of nursing unit, nursing care left undone, and nurse-reported quality of care (*n* = 230).

Variables	Range	m ± SD
Team effectiveness (total)	(1–5)	3.80 ± 0.55
Leadership of the head nurse	(1–5)	3.94 ± 0.78
Cohesion	(1–5)	3.90 ± 0.61
Job satisfaction	(1–5)	3.73 ± 0.65
Competency of nurses	(1–5)	3.94 ± 0.64
Work productivity	(1–5)	3.76 ± 0.67
Coordination	(1–5)	3.55 ± 0.72
Nursing care left undone	(0–13)	1.43 ± 1.78
Nurse-reported quality of care	(1–5)	3.75 ± 0.71

**Table 3 healthcare-11-01380-t003:** Relationship among team effectiveness of the nursing unit, nursing care left undone, and nurse-reported quality of care (*n* = 230).

Variables	Nursing Care Left Undone	Nurse-Reported Quality of Care
Team effectiveness (total)	−0.23 **	0.59 **
Leadership of the head nurse	−0.14 *	0.43 **
Cohesion	−0.16 *	0.45 **
Job satisfaction	−0.15 *	0.44 **
Competency of nurses	−0.21 **	0.54 **
Work productivity	−0.23 **	0.55 **
Coordination	−0.25 **	0.48 **
Nursing care left undone		−0.31 **

* *p* < 0.05, ** *p* < 0.001.

**Table 4 healthcare-11-01380-t004:** Factors influencing nursing care left undone and nurse-reported quality of care (*n* = 230).

Variables (Reference)	Nursing Care Left Undone	Nurse-Reported Quality of Care
	β	β
Years of nursing experience in the current unit	−0.02	0.02
Integrated nursing care service (general nursing care service)	−0.17 **	0.12 *
Leadership of the head nurse	0.01	0.12
Cohesion	0.03	−0.02
Job satisfaction	0.07	0.05
Competency of nurses	−0.16	0.26 **
Work productivity	−0.02	0.20 *
Coordination	−0.22 *	0.08
Nursing care left undone		−0.15 **
F(p)	3.17 **	16.83 **
Adjusted R^2^	0.07	0.38

* *p* < 0.05, ** *p* < 0.001.

## Data Availability

The original contributions presented in the study are included in the article, further inquiries can be directed to the corresponding author.

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
