# Peer review of "Influence of the Team Effectiveness of Nursing Units on Nursing Care Left Undone and Nurse-Reported Quality of Care"

_healthcare, 2023, doi:10.3390/healthcare11101380_

Round 1

Reviewer 1 Report

Thank you for the opportunity to review an interesting study.

It looks like a well-planned study, but the overall flow needs revision.

1. Abstract : Please add the method and timing of the survey.

2. Instruction

- Line 29-31 (According to the Intergrated Team Efectiveness Model ~) : This sentence is a quotation, so please attach a reference.

- This study explored the impact of the sub-domains of the Team Effectiveness of Nursing Unit on Nursing Care Left Undone and Nurse-Reported Quality of Care, respectively. However, the description of the sub-domains of the Team Effectiveness of Nursing Unit is somewhat lacking. Please add this part.

- Also, the significance of Nursing Care Left Undone functioning as a dependent variable along with Nurse-Reported Quality of Care is insufficient. How does Nurse-Reported Quality of Care function in healthcare? What does it mean? Please be kinder to describe this part.

3. Methods: The research method is appropriately described as to what needs to be described.

4. Result: 

- In the description of n (%) in Table 1, put a space between n and parentheses.

- Table 4: Adj R2 -> Adj. R2 (2 is superscript), add the full term of Adj at the bottom of the table.

5. Discussion

- The contents of Line 201-211 seem to be more suitable for Instruction. It is recommended to use the contents to reinforce the Instruction.

- It is recommended that the contents of Lines 222-237 be incorporated into paragraphs that follow.

- Likewise, it would be good to incorporate paragraphs on lines 238-259 as well.

Author Response

We wish to thank you for your thoughtful comments and valuable feedback on the manuscript entitled “Influence of the Team Effectiveness of Nursing Unit on Nursing Care Left Undone and Nurse-Reported Quality of Care.” We would like to resubmit the revised manuscript for publication in the Healthcare.

We have tried to revise the manuscript according to your suggestions and rewrote or rephrased sections to improve clarity. For your convenience, we have used red font for the revisions. Please find the following revisions according to reviewer’s comments.

Further, I believe that this revised paper will be of interest to the readership of the Healthcare. Thank you for your consideration. I look forward to hearing from you.

Reviewer 2 Report

Thank you very much for the opportunity to review your article.

The article is very interesting, and there will be some other research trips such as cost effectiveness, and quality of life pofesional or job satisfaction of these nurses.

For my part it is a very interesting article.

Just to say that the title is difficult to understand, perhaps it would be advisable to modify and adjust it to the aim of the study.

The introduction is correct, and puts in background. the methodology, could be improved, as for example as they filled in the questions, whether it was online or on paper, as they were registered, and above all consent.( although they explain that I pass the approval of the committee at the end of everything).

at a statistical level as it is a descriptive study is correct.

the bibliography if I would look for more recent, or any revision if there were.

congratulations on the job.

Author Response

(The authors gave the same response as above.)

Reviewer 3 Report

I am concerned about the rationale for using the Integrated team effectiveness model (ITEM) from 2006. There are many more recent team effectiveness models and validated team effectiveness tools to use. I looked at the 2006 reference, and the figure is quite complex. I am not sure why you chose to use the variables you used, based on the ITEM model. More explanation is needed, therefore, for the use of the ITEM model and the outcomes variables. 

I question your definition of team and your decision to use the nursing unit as a proxy for "team." I recommend using the care delivery mode as the unit of analysis (e.g., team-based versus total care or primary care). Teams consider themselves teams. Do nurses consider their nursing unit a team? 

Context is important, and descriptions of differences between the general wards and integrated wards is not provided. It sounds as if your research question really focused on differences in team performance and outcomes between general wards and integrated wards. I was also surprised that the average number of patients per nurse was 2 patients, which sounds like critical care. 

Author Response

(The authors gave the same response as above.)

Round 2

Reviewer 3 Report

Thank you for your revisions in response to reviewer requests. The manuscript reads very well, and you have some interesting findings for the journal's readership.